Prognostic value of serum Mrp 8/14 in sepsis-induced acute respiratory distress syndrome patients: a retrospective cohort study

Sun Caizhi 1 2
Xie Yongpeng 1
Zhu Chenchen 2
Guo Lei 2
Xu Bowen 2
Qin Haidong 2
Li Xiaomin 1 lyglxmsj@163.com
1 Department of Emergency Medicine, Lianyungang Clinical College, Nanjing Medical University , Lianyungang, Jiangsu , China
2 Department of Emergency Medicine, Nanjing First Hospital, Nanjing Medical University , Nanjing, Jiangsu , China
Upadhyay Rohit
Electronic publication date: 2024 Dec 13
Publication date: 2024
Volume: 12
Electronic Location ID: e18718
Received 2024 Aug 8; Accepted 2024 Nov 25
Copyright: © 2024 Sun et al.
Copyright year: 2024
Copyright holder: Sun et al.
License: This is an open access article distributed under the terms of the Creative Commons Attribution License, which permits unrestricted use, distribution, reproduction and adaptation in any medium and for any purpose provided that it is properly attributed. For attribution, the original author(s), title, publication source (PeerJ) and either DOI or URL of the article must be cited.
License URL: https://creativecommons.org/licenses/by/4.0/

Keywords: Mrp 8/14, Sepsis, Acute respiratory distress dyndrome, Prognosis, Mortality

Funding: Social Development Project of Department of Science and Technology of Jiangsu Province BE2020670 Nanjing Medical Science and Technique Development Foundation Project YKK22115, YKK20114 This work was supported by the Social Development Project of Department of Science and Technology of Jiangsu Province (No. BE2020670) and the Nanjing Medical Science and Technique Development Foundation Project (Nos. YKK22115, YKK20114), China. The funders had no role in study design, data collection and analysis, decision to publish, or preparation of the manuscript.

==============================
Background

Mrp 8/14 is abundantly secreted by activated neutrophils during infection and inflammation. However, its prognostic value in acute respiratory distress dyndrome (ARDS) induced by sepsis is poorly understood. Our aim was to investigate the relationship between serum Mrp 8/14 and the prognosis in sepsis-induced ARDS patients admitted to the intensive care unit (ICU).

Methods

Serum Mrp 8/14 concentrations were analyzed in 118 ARDS patients induced by sepsis included in the analytical study. Patients were enrolled upon admission to the ICU of Nanjing Hospital affiliated to Nanjing Medical University. The baseline information and clinical outcomes were obtained. Patients were divided into survivor group and non-survivor group according to whether they died during ICU hospitalization.

Results

The serum Mrp 8/14 levels were significantly increased in the non-survivor group compared to the survivor group (P < 0.05). Logistic regression analysis showed that serum Mrp 8/14, albumin and APACHE II were the independent factors for predicting the prognosis of sepsis-induced ARDS during ICU hospitalization after adjustment. Additionally, the area under the receiver operating characteristic curve for Mrp 8/14 combined with albumin was associated with ICU mortality and was higher than that of Mrp 8/14, albumin, APACHE II and Mrp 8/14 combined with APACHE II (all P < 0.05). A nomogram was constructed to predict ICU mortality and the c-indexes of predictive accuracy was 0.830 in the cohort (P < 0.05).

Conclusions

The serum Mrp 8/14 upon ICU admission in septic patients may be useful for predicting mortality in sepsis-induced ARDS patients during ICU hospitalization.

Introduction

Sepsis refers to a multi-organ dysfunction caused by a dysregulated immune and inflammatory response to infection, which is one of the most prevalent complications in the intensive care unit (ICU) (Hollenberg & Singer, 2021). As a reservoir of pathogens in the body, the lung is frequently the initial target organ susceptible to damage during sepsis. Moreover, the impaired exchange of oxygen and nutrients in the lung tissue is regarded as the most formidable complication of septic lung injury. Severe lung injury induced by sepsis can ultimately progress to the development of acute respiratory distress syndrome (ARDS) and result in profound respiratory failure (Hwang et al., 2019).

ARDS, as an acute, diffuse, inflammatory form of lung injury, represents a life-threatening condition observed in critically ill patients. According to a recent report, the incidence of ARDS in the United States ranges from 64.2 to 78.9 cases per 100,000 person-years (Diamond et al., 2024). It is noteworthy that despite recent advancements in critical care therapeutic interventions, the hospital mortality rate associated with sepsis-induced ARDS remains significantly high (Auriemma et al., 2020). Furthermore, we cannot ignore that a considerable proportion of individuals who have survived ARDS encounter functional limitations, such as muscular weakness, respiratory symptoms, anxiety, depressive tendencies, and cognitive impairments and so on (Hodgson et al., 2022). In addition, the comprehensive respiratory mechanics parameters elastic power and mechanical power were associated with the clinical prognosis in ARDS patients (Xie et al., 2021, 2023). However, all the enrolled ARDS patients in the conducted studies were subjected to invasive mechanical ventilation, thereby limiting the ability to assess the prognostic value of elastic power and mechanical power in ARDS patients under varying oxygen therapy conditions. Therefore, it is required to identify an enhanced biomarker that can accurately predict the clinical outcome of ARDS in patients with sepsis during ICU hospitalization.

It is widely acknowledged that the activation of neutrophils and inflammatory response play a pivotal role in the development of ARDS among critically ill patients with sepsis (Zhang et al., 2011). The protein myeloid-related proteins 8 and 14 (MRP 8/14), also known as calprotectin or S100 calcium binding protein A8 and A9 (S100 A8/A9), represents a potent pro-infammatory alarmin stored in significant quantities within neutrophils and released upon activation (Pruenster et al., 2016). Recently, there has been a growing concern regarding the correlation between calprotectin and sepsis as well as its associated complications. Research has demonstrated that the initial serum concentrations of calprotectin exhibited a robust diagnostic efficacy for bacterial sepsis (Bartáková et al., 2019). A subsequent clinical study also showed that calprotectin was significantly increased in critically ill patients with sepsis and that higher calprotectin concentrations at ICU admission predict mortality risk (Wirtz et al., 2020). Furthermore, recent studies have also indicated an association between S100A9 and COVID-19-induced pneumonia as well as ARDS (Lee et al., 2021; Kassianidis et al., 2022), suggesting a potential significant impact of S100A9 on ARDS induced by COVID-19. Similarly, deficiency or inhibition of S100A8/A9 in mice conferred protection against lung injury development in endotoxin-induced sepsis and improve the survival (Su et al., 2022). It is worth noting that we cannot ignore the physiological role and function of MRP 8/14 in the immune system (Romand et al., 2019; Inciarte-Mundo, Frade-Sosa & Sanmartí, 2022). Tissue damage triggers the release of damage-associated molecular pattern molecules and pro-inflammatory cytokines. In response to the upregulation of pattern recognition receptors, neutrophils are rapidly recruited (Drury et al., 2021; Kim et al., 2021) and secrete a diverse array of chemokines, cytokines and leukotrienes, including MRP 8/14 during the inflammatory response. In the text of autoimmunity, MRP 8/14 may serve as a link between inflammation and the adaptive immune response, thereby contributing to the activation of autoreactive CD8+ T cells. Additionally, in association with CD40/CD40 ligand signalling, it leads to the breakdown of T cell tolerance (Loser et al., 2010). MRP 8/14 may play a causative role in the pathogenesis of severe sepsis. In addition, our previous study has revealed that serum Mrp 8/14 exhibits a strong predictive value in the occurrence of ARDS among septic patients (Sun et al., 2024). However, whether MRP 8/14 could be used as a potential prognostic indicator of ARDS patients induced by sepsis during ICU hospitalization has not been previously investigated.

Therefore, the aim of the present study was to investigate whether higher serum Mrp 8/14 level measured shortly after hospitalization (usually within 24 h) would be associated with worse clinical outcomes in ARDS induced by sepsis patients during ICU hospitalization. Additionally, while numerous reports have documented the successful establishment of nomograms for disease prognostics, there remains a paucity of nomograms specifically designed to predict ARDS outcomes. Therefore, we sought to assess the predictive value of a novel nomogram based on Mrp 8/14 in sepsis-induced ARDS patients during ICU hospitalization.

Patients and Methods

Study design and participants

This present study is a single-center, retrospective cohort investigation that enrolled patients who were admitted to ICU of Nanjing Hospital affiliated to Nanjing Medical University from August 2021 to July 2023. The enrolled ARDS patients were obtained from our previous study (Sun et al., 2024) and divided into survivor group and non-survivor group according to whether they died during ICU hospitalization. The survivor group was defined as ARDS patients who improved after treatment and were discharged or transferred from ICU to the general ward. The non-survivor group was defined as ARDS patients who died after treatment during ICU hospitalization.

This study protocol was reviewed and approved by the Institutional Review Board of Nanjing Hospital Affiliated to Nanjing Medical University (IRB no: KY20201102-03) and was conducted in accordance with the Declaration of Helsinki. All participants were over 18 years old and signed informed consent. If the patients were intubated or the condition was too critical to obtain consent upon admission, guardians or parents provided their signature on the consent forms in the present study. Upon regaining consciousness, the patients would be informed and provided additional informed consent.

Clinical data evaluation and serum Mrp 8/14 detection

The baseline information were collected and serum Mrp 8/14 levels were detected as previously described in the previous studies (Sun et al., 2024, 2023). The following laboratory indicators based from the same collection date were also assessed: white blood cell count (WBC), neutrophil count, procalcitonin (PCT) level, C-reactive protein (CRP) level, interleukin-6 (IL-6), platelet, albumin, sodium, calcium, creatinine, blood urea nitrogen (Bun) and the proporation of patients with extracorporeal membrane oxygenation (ECMO) or prone. Hypernatremia was defined as [Na+] > 145 mmol/L. In addition, arterial oxygen partial pressure/inspired oxygen fraction (PaO2/FiO2) was utilized to classify the severity of ARDS patients. “mild” was defined as a PaO2/FiO2 ratio ranging from 200 to 299 mmHg, while “moderate” referred to a PaO2/FiO2 ratio ranging from 100 to 199 mmHg, and “severe” denoted a PaO2/FiO2 ratio below 100 mmHg. The comorbid conditions such as hypertension (67.80%, 80/118), coronary heart disease (29.66%, 35/118), cerebrovascular disease (30.51%, 36/118) and diabetes mellitus (47.46%, 56/118) were collected. The sites of infection in sepsis-induced ARDS patients leading to ICU admission such as respiratory infection (55.08%, 65/118), urinary tract infection (10.17%, 12/118), bloodstream infection (8.47%, 10/118), abdominal cavity infection (19.49%, 23/118) and others (6.78%, 8/118) were recorded.

In addition, the lung ultrasound score (LUS) was obtained upon diagnosis of ARDS in septic patients during their ICU hospitalization. The chest wall of each enrolled patient was divided into 12 areas. The LU scores were calculated as the cumulative sum of estimated values for each site. Each district was assessed based on the following criteria: presence of A lines or less than two isolated B lines denoted as 0, presence of discrete B lines denoted as 1, presence of fused B lines denoted as 2, and presence of lung consolidation area denoted as 3. Higher scores indicated bigger lung damage.

Statistical analysis

Continuous variables were tested for normal distribution using the Kolmogorov-Smirnov test. Continuous data were expressed as means ± standard deviation (SD) if they met the normal distribution, and the independent-sample t-test was used for inter-group comparison. Non-normally distributed continuous data were expressed as the median and interquartile range (IQR). The Mann-Whitney U-test was used for comparison. Qualitative/categorical variables were compared between groups using the chi-square test for continuity, and the results are shown as numbers and percentages. Univariable analysis was used to investigate the prognostic significance of continuous and categorical measures in sepsis-induced ARDS patients. Significant predictors in the univariable analysis were included in a multivariable regression model to determine independent predictors. The ability of serum Mrp 8/14 to predict the ICU mortality was analyzed by the receiver operating characteristic (ROC) curves. Additionally, a nomogram and calibration curve was established using R for Windows with the package rms. The discriminatory ability of the nomogram was evaluated using Harrell’s c-index. Calibration was used to compare the actual ICU mortality with the predicted probability predicted by the curve. Statistical analysis was performed with SPSS software (Version 27; SPSS Inc., Chicago, IL, USA). Data in the present study were collected and analyzed following the previously described methods in Sun et al. (2024), Deng et al. (2018). A P value less than 0.05 was considered statistically significant.

Results

Baseline of patients’ characteristics

There were 164 ARDS patients induced by sepsis who met the criteria of Sepsis and Septic Shock 3.0 and Berlin definition. A total of 46 patients were excluded according to the following criteria: lack of data (n = 9), age <18 years (n = 1), less than 24 h stay or more than 30 days (n = 16), malignancies, immunodeficiency or pregnancy (n = 8) and history of asthma, COPD or interstitial lung disease (n = 12). A total of 118 patients met the inclusion criteria, including 64 patients in survivor group and 54 patients in non-survivor group (Fig. 1). The median age in survivor group was 69.77 ± 9.28 years with 59.38% being men and the non-survivor group was 72.48 ± 7.78 years with 64.81% being men. Compared to the survivor group, patients in the non-survivor group demonstrated significantly elevated APACHE II and SOFA scores, as well as increased levels of PCT, Bun, hypernatremia, LUS and proportion of severe ARDS patients, while the albumin level was lower (all P < 0.05). Moreover, no significant difference was found in other baseline characteristics and the use of ECMO or prone treatment in ARDS patients between the two groups (P > 0.05) (Table 1).

Figure 1 Flow chart of the study patients to illustrate study screening and ecruitment.

Abbreviations: COPD, chronic obstructive pulmonary disease; ARDS, Acute respiratory distress syndrome.

Table 1 Baseline clinical and laboratory characteristics of the study population.

Patients characteristics	Survivors (n = 64)	Non-survivors (n = 54)	χ2/Z/t	P	
Sex (male, n (%))	38 (59.38)	35 (64.81)	0.367	0.544	
Age, years, mean ± SD	69.77 ± 9.28	72.48 ± 7.78	−1.705	0.091	
Infection sites leading to ICU admission, n (%)			0.322	0.988	
Respiratory infection	36	29	0.077	0.782	
Urinary tract infection	7	5	0.090	0.764	
Bloodstream infections	5	5	0.079	0.779	
Abdominal cavity infection	13	10	0.060	0.806	
Others	5	3	0.014	0.906	
Past medical history, n (%)			0.515	0.916	
Coronary heart disease	20	15	0.169	0.681	
Hypertension	45	35	0.405	0.524	
Diabetes mellitus	30	26	0.019	0.890	
Cerebrovascular disease	22	14	0.986	0.321	
Severity of ARDS, n (%)			4.738	0.094	
Mild	24 (37.50)	16 (29.63)	0.810	0.368	
Moderate	34 (53.13)	25 (46.30)	0.546	0.460	
Severe	6 (9.37)	13 (24.07)	4.684	0.030	
Laboratory indicators	
WBC, ×109/L, mean ± SD	14.39 ± 6.09	13.67 ± 4.85	0.732	0.466	
Neutrophil, ×109/L, mean ± SD	11.50 ± 5.12	10.26 ± 3.95	1.453	0.149	
CRP, mg/L, mean ± SD	115.58 ± 77.63	120.96 ± 83.47	−0.362	0.718	
PCT, ng/mL, median (IQR)	13.56 (3.85, 41.49)	31.00 (16.81, 53.07)	−3.079	0.002	
IL-6, pg/mL, median (IQR)	1,162.31 (126.62, 9,269.45)	5,215.70 (295.25, 10,190.28)	−1.631	0.103	
Platelet, ×109/L, mean ± SD	169.42 ± 83.24	150.76 ± 98.04	1.118	0.266	
Albumin, g/L, mean ± SD	30.42 ± 4.52	27.35 ± 4.30	3.749	0.000	
ALT, U/L, median (IQR)	25.00 (12.25, 51.00)	19.50 (10.00, 35.50)	−1.430	0.153	
AST, U/L, median (IQR)	29.50 (18.00, 96.00)	33.00 (19.00, 63.50)	−0.600	0.549	
Creatinine, umol/L, median (IQR)	122.95 (72.08, 78.25)	139.40 (78.50, 391.43)	−1.602	0.109	
Bun, mol/L, median (IQR)	11.13 (6.54, 9.08)	16.05 (7.86, 30.53)	−2.639	0.008	
APACHE II score, mean ± SD	25.52 ± 4.74	27.80 ± 4.81	−2.585	0.011	
SOFA score, mean ± SD	9.89 ± 4.00	12.00 ± 3.45	−3.034	0.003	
Calcium, mmol/L, median (IQR)	1.14 (1.07, 1.23)	1.15 (1.07, 1.23)	−0.459	0.621	
Sodium, mmol/L, median (IQR)	137.68 (133.71, 142.95)	137.50 (131.78, 157.22)	−0.686	0.493	
Hypernatremia, n (%)	10 (15.63)	20 (37.04)	7.082	0.008	
LUS, mean ± SD	16.61 ± 5.21	20.48 ± 5.32	−3.986	0.000	
Mrp 8/14, ug/ml, median (IQR)	12.89 (8.74, 17.83)	21.04 (10.92, 28.62)	−4.889	0.000	
Patients with ECMO, n (%)	12 (18.75%)	16 (29.63%)	1.916	0.166	
Patients with prone, n (%)	6 (9.38%)	8 (14.81%)	0.829	0.363	
Note:

Abbreviations: APACHE II score, Acute physiology and chronic health evaluation II score; WBC, White blood cell; PCT, Procalcitonin; SOFA score, Sequential Organ Failure Assessment score; IL-6, Interleukin-6; LUS, Lung ultrasound; Bun, Blood urea nitrogen; ALT, Alanine aminotransferase; AST, Aspartate aminotransferase; Mrp 8/14, Myeloid-related proteins 8 and 14; ECMO, Extracorporeal membrane oxygenation; IQR, Inter quartile range; SD, Standard deviation. P value below 0.05 indicate statistical significance.

Level of serum Mrp 8/14 in sepsis-induced ARDS patients

All serum samples were collected from 118 ARDS patients within 24 h of ICU admission. Median (IQR) serum Mrp 8/14 levels in the non-survivor group were 21.04 (10.92, 28.62) ug/ml, which significantly exceeded the level of Mrp 8/14 in the survivor group at 12.89 (8.74, 17.83) ug/ml (P = 0.000) (Table 1).

Comparison of clinical outcomes

The clinical outcome of ARDS patients induecd by sepsis was evaluated based on the following parameters presented in Table 2, including the length of ICU stay, hospitalization expenses in ICU, shock rate, the duration of mechanical ventilation and renal replacement therapy. The results showed that hospitalization expenses in ICU, shock rate, duration of mechanical ventilation and renal replacement therapy were significantly higher in non-survivor group than that in survivor group (10.85 (6.75, 16.44) vs. 7.84 (5.93, 10.45), P = 0.004; χ2 = 4.543, P = 0.033; χ2 = −2.986, P = 0.003; χ2 = 4.074, P = 0.044). However, the length of ICU stay exhibit no significant differences between the two groups (P > 0.05).

Table 2 Clinical outcomes of the two groups.

Outcomes	Survivors (n = 64)	Non-survivors (n = 54)	χ2/Z	P	
Renal replacement therapy, n (%)	15 (23.44)	22 (40.74)	4.074	0.044	
Shock rate, n (%)	38 (59.38)	42 (77.78)	4.543	0.033	
Duration of mechanical ventilation, days, median (IQR)	4.00 (2.00–7.00)	6.00 (4.00, 11.00)	−2.986	0.003	
Length of ICU stay, days, median (IQR)	10.00 (7.00, 13.00)	9.50 (5.00, 15.00)	−0.228	0.820	
Hospitalization expenses in ICU, median (IQR), (×10,000 yuan)	7.84 (5.93, 10.45)	10.85 (6.75, 16.44)	−2.879	0.004	
Note:

Abbreviations: ICU, Intensive care unit; IQR, Inter quartile range. P value below 0.05 indicate statistical significance.

Factors associated with the mortality of sepsis-induced ARDS during ICU hospitalization

The Forest plot of logistic regression analysis suggested that APACHE II, albumin and Mrp 8/14 were the independent factors for the mortality of sepsis-induced ARDS during ICU hospitalization (APACHE II: OR = 1.136, 95% CI = [1.019–1.267], P = 0.022; albumin: OR = 0.839, 95% CI = [0.742–0.948], P = 0.005; Mrp 8/14: OR = 1.084, 95% CI = [1.009–1.165], P = 0.027), as illustrated in Fig. 2.

Figure 2 Forest plot of logistic regression analysis for ICU mortality.

The logistic regression analysis showed Mrp 8/14, LUS, Albumin and APACHE II were the independent fator for predicting the ICU mortality of ARDS induced by sepsis. Abbreviations: PCT, Procalcitonin; LUS, Lung ultrasound score; Bun, Blood urea nitrogen; ICU, Intensive care unit; Mrp 8/14, Myeloid-related proteins 8 and 14; ARDS, Acute respiratory distress syndrome; APACHE II score, Acute physiology and chronic health evaluation II score; SOFA score, Sequential Organ Failure Assessment score; CI, Confidence interval; OR, Odds ratio. P value below 0.05 indicate statistical significance.

Predictive value of Mrp 8/14 for ICU mortality of ARDS patients

To investigate whether serum Mrp 8/14 is useful in predicting the mortality of sepsis-induced ARDS during ICU hospitalization, the ROC analysis was conducted using clinical and laboratory indicators. As shown in Fig. 3 and Table 3, the expression of serum Mrp 8/14 had a certain value of being used as a prognostic indicator for ICU mortality of patients with ARDS induced by sepsis (AUC 0.762, 95% CI [0.674–0.850], P = 0.000). In addition, the ROC also showed the clinical significance of albumin and APACHE II score in predicting the ICU mortality of ARDS patients (Albumin: AUC 0.694, 95% CI [0.599–0.788], P = 0.000; APACHE II: AUC 0.617, 95% CI [0.517–0.718], P = 0.028). Although these three indicators exhibited a certain degree of value in distinguishing the mortality outcome of patients with ARDS during ICU hospitalization, they had a relatively low specificity or sensitivity (Table 3). Therefore, the combined prediction was analyzed by employing the ROC curve. The results demonstrated that Mrp 8/14 combined with albumin had higher accuracy in screening the prognosis of patients with ARDS induced by sepsis (AUC 0.810, 95% CI [0.733–0.887], P = 0.000).

Figure 3 Receiver operating characteristic (ROC) curve analysis of Mrp 8/14 for predicting the mortality of ARDS induced by sepsis during ICU hospitalization.

Abbreviations: ICU, Intensive care unit; Mrp 8/14, Myeloid-related proteins 8 and 14; ARDS, Acute respiratory distress syndrome; APACHE II score, Acute physiology and chronic health evaluation II score; AUC, Area under the curve. P value below 0.05 indicate statistical significance.

Table 3 The performance of Mrp 8/14 for predicting ICU mortality in ARDS induced by sepsis.

Indicators	Cut-off	AUC (95% CI)	Sensitivity	Specificity	Youden	P	
Mrp 8/14	15.61	0.762 [0.674–0.850]	66.67	79.69	0.464	0.000	
Albumin	29.95	0.694 [0.599–0.788]	77.80	54.70	0.325	0.000	
APACHE II	26.5	0.617 [0.517–0.718]	57.40	60.90	0.183	0.028	
Mrp 8/14 combined with Albumin	–	0.810 [0.733–0.887]	61.10	87.50	0.486	0.000	
Mrp 8/14 combined with APACHE II	–	0.791 [0.709–0.872]	75.90	71.90	0.478	0.000	
Note:

Abbreviations: APACHE II score, Acute physiology and chronic health evaluation II score; ICU, Intensive care unit; Mrp 8/14, Myeloid-related proteins 8 and 14; ARDS, Acute respiratory distress syndrome; CI, Confidence interval; AUC, Area under the curve. P value below 0.05 indicate statistical significance.

New prognostic model

To further evaluate the predictive ability of serum Mrp 8/14 in sepsis-induced ARDS patients, we used a nomogram based on the findings from the multivariate regression analyses to predict ICU mortality. The multivariate regression analysis confirmed that APACHE II, albumin and Mrp 8/14 were significantly correlated with ICU mortality in the cohort patients. Therefore, the indicators of APACHE II, albumin and Mrp 8/14 were included in the predictive model for the cohort (Fig. 4A). R software was used with stepwise Cox regression analyses. The nomogram for the cohort showed that APACHE II, albumin and Mrp 8/14 were indicators of a poor prognosis. These findings were consistent with the previous results obtained from the multivariate analyses (Fig. 2). Furthermore, calibration curves for the predictive model in the cohort indicated that the predicted ICU mortality values were similar to the actual ICU mortality (Fig. 4B). We then assessed the predictive accuracy of the prognostic model; the Harrell’s c-index values for the nomogram in the cohort was 0.830 (P < 0.05).

Figure 4 Nomogram of the study population to predict ICU mortality in ARDS induced by sepsis during ICU hospitalization.

The nomogram is used by summing the points assigned to the corresponding factors, which are presented at the top of the scale. The total is used to predict the ICU probability of mortality in the highest scale. The c-indexes values for the cohort (A) is 0.830. Calibration curve for ICU mortality, which is representative of predictive accuracy, for the cohort (B). The 45-degree reference line represents a perfect match between predicted and observed values. Abbreviations: ICU, Intensive care unit; Mrp 8/14, Myeloid-related proteins 8 and 14; ARDS, Acute respiratory distress syndrome; APACHE II score, Acute physiology and chronic health evaluation II score.

Discussion

To our knowledge, this study represents the first investigation into the prognostic value of serum Mrp 8/14 in patients with sepsis-indued ARDS patients during ICU hospitalization. A previous study has reported the correlation between serum Mrp 8/14 levels and mortality in septic patients, while the downregulation or deficiency of Mrp 8/14 expression has shown potential for improving lung injury in septic mice (Zhao et al., 2021). In the present study, we observed a novel finding in sepsis-induced ARDS patients, indicating that elevated serum Mrp 8/14 levels were significantly associated with an increased risk of mortality during ICU stay, thereby strengthening the existing evidence for the crucial role of serum Mrp 8/14 in the pathobiology and prognosis of ARDS among critically ill patients with sepsis.

Studies have demonstrated that excessive neutrophilic inflammation contributed to the pathogenesis of lipopolysaccharide-induced lung injury (Wang et al., 2019). As a protein associated with inflammatory responses, Mrp 8/14 is abundantly released and plays a crucial role in recruiting leukocytes and facilitating cytokine secretion upon infection, thereby exacerbating inflammatory responses and promoting tissue damage (Kovačić et al., 2018; Revenstorff et al., 2022). In our study, we observed a clear detectable difference in serum Mrp 8/14 expression level between the survivor and non-survivor group of ARDS patients, which further demonstrated that the levels of serum Mrp 8/14 was significantly elevated in patients with ARDS (Kassianidis et al., 2022). Moreover, the data in the present study showed that the hospital mortality of sepsis-induced ARDS admitted to the ICU was 45.76% (54/118), which is consistent with the previous studies that the mortality associated with ARDS is estimated to be approximately 40–50% (Hu, Hao & Tang, 2020). Further analysis revealed that respiratory infection accounted for 55.08% (65/118) of the common causes of sepsis-induced ARDS, which aligns with the study indicating that pneumonia was the cause of nearly 60% ARDS cases in the LUNG-SAFE cohort (Bellani et al., 2016).

However, we did not observe a significant increase in WBC and neutrophil count among the non-survivor ARDS patients compared to those who survived, despite these two indices being significantly higher than the normal range. In fact, relying solely on a single blood cell parameter such as WBC or neutrophil counts may not be the optimal approach for accurately identifying sepsis or ARDS due to its susceptibility to various non-infectious factors when administered with corticosteroids, catecholamines, and critical illness stress (Ho et al., 2023). Moreover, in the context of severe infection, particularly if the ARDS patients have compromised immune function, these parameters may even exhibit a marked decrease (Murray et al., 2007). As a pro-inflammatory cytokine, IL-6 has been reported to be a prominent prognostic marker for assessing the severity of sepsis, and its clinical significance has been extensively investigated in various septic conditions through numberous studies (Vago et al., 2023; Ma et al., 2016). Nevertheless, the results of our study here revealed no significance of serum IL-6 levels in evaluating the prognosis of patients with sepsis-induced ARDS. A preliminary analysis considered that the difference may relate to the overall higher inflammatory response triggered by an active polymicrobial infection.

Albumin, as a significant protein reflecting the body’s nutritional status, plays an important role in various physiological procedure (Cabrerizo et al., 2015). LUS is an important noninvasive technique for evaluating the water-to-gas ratio in lung and reflecting pulmonary injury through semiquantitative scoring (Smit et al., 2021). In the present study, we observed a significant decrease in serum albumin levels in the non-survivor group compared to the survivor group, while conversely, the LUS exhibited an opposite trend. Our research findings indicated that decreased levels of albumin upon admission to ICU were associated with an unfavorable prognosis in sepsis-induced ARDS, thereby further providing evidence for a correlation between lower albumin levels and poorer clinical outcomes in infectious diseases (Ding et al., 2022). Several potential mechanisms may account for the reduction in serum albumin levels observed in septic ARDS patients. First, the elevated systemic inflammatory factors may potentially impair the function of vascular endothelium and increase capillary vessels permeability. Subsequently, extravasation of albumin from the vasculature may lead to a reduction in plasma albumin levels (Han et al., 2022). Moreover, inflammation impairs the renal function and induce albumin leakage (Liang et al., 2023; Qin et al., 2022). Third, sepsis-induced ARDS patients commonly present with concurrent gastrointestinal dysfunction, which impedes nutrient absorption and causes malnutrition status (Hsieh et al., 2020).

Further multivariable analysis suggested that the mortality of sepsis-induced ARDS patients admitted to ICU was associated with APACHE II score, Mrp 8/14 and albumin levels. ROC analysis demonstrated that serum Mrp 8/14 showed a higher specificity of 79.69% as a single indicator compared to albumin and APACHE II. When combined with albumin, the predictive specificity reached an impressive 87.50% and improved the predictive value of ICU mortality in sepsis-induced ARDS patients. The nomogram, as a visual and widely accepted approach for predicting disease prognosis based on various clinical characteristics, has demonstrated superior prognostic predictive accuracy compared to traditional tumor staging systems in malignancies (Wang et al., 2013). In the present study, we further attempted to develop a prognostic nomogram for the prediction of ICU mortality in sepsis-induced ARDS patients. Based on our nomogram, Mrp 8/14 was included in the final model through a stepwise algorithm. The predictive effect of the nomogram was well explained by the calibration curve in the cohort (Figs. 4B). The results from nomogram showed that APACHE II and albumin were also predictors of a poor prognosis in patients with ARDS induced by sepsis. Therefore, Mrp 8/14 should be considered when predicting the prognosis of sepsis-induced ARDS patients during ICU hospitalization.

It is noteworthy that there was no significant difference in the duration of ICU stay between the two groups, despite higher expenses associated with ICU hospitalization observed in the non-survivors group. The above phenomenon may be attributed to prolonged periods of renal replacement therapy and mechanical ventilation, increased dosages of vasoactive drugs, as well as a greater severity of illness and poorer treatment response among non-survivors. In addition, although no significant difference in serum sodium levels was observed between the surviving ARDS patients and those who deceased in the present study, the prevalence of hypernatremia was significantly higher in the non-survivor group compared to that of the survivor group (37.04% vs. 15.63%), which was consistent with the study indicating that hypernatremia upon ICU admission is associated with a poorer prognosis and increased mortality in sepsis patients (Lin et al., 2022).

In addition, we cannot ignore that the inclusion of ARDS patients in the present study was still based on the Berlin definition (ARDS Definition Task Force et al., 2012) rather than the most recent New Global Definition (Matthay et al., 2024). The New Global Definition is widely acknowledged as an extension of the Berlin definition of ARDS, aiming primarily to enhance the diagnostic feasibility of ARDS, particularly in settings with limited resources. However, the diagnosis of ARDS patients admitted to the ICU primarily relies on comprehensive clinical presentation evaluation, PaO2/FiO2 ratio and identification of new infiltrates on chest imaging. The diagnostic approach remains consistent with the New Global Definition. Therefore, the results of the present study based on the Berlin definition remained unaffected by the latest diagnostic criteria for ARDS.

However, the limitations of the present study need to be acknowledged when interpreting these findings. First, to ensure the certainty of ARDS diagnosis, we excluded the patients who had been diagnosed with ARDS prior to admission to ICU, which may potentially affect the prognostic value of MRP 8/14 in patients with sepsis-induced ARDS to a certain extent. Additionally, the findings presented here are limited in their generalizability to ARDS patients induced by factors other than sepsis. Validation for for ARDS induced by alternative etiological factors is also necessary. Finally, Mrp 8/14 was included in the final model after regression analysis. Nevertheless, the dynamic changes in serum Mrp8/14 during treatment and their potential impact on the prognosis of sepsis-induced ARDS patients during ICU hospitalization remain unconfirmed.

In conclusion, the data presented validates that Mrp 8/14, as an objective parameter, serves as a individualized prognostic indicator for mortality during ICU hospitalization. Mrp 8/14, in combination with APACHE II and albumin, might improve the predictive accuracy for ICU mortality after sepsis-induced ARDS patients. The present study had yielded valuable insights into the issue; however, further larger prospective studies are warranted to validate the current findings. Considering all aspects, we should pay more attention to the serum Mrp 8/14 levels upon ICU admission in sepsis-induced ARDS patients during ICU hospitalization.

Supplemental Information

Supplemental Information 1 Raw data.

Abbreviations

APACHE II score Acute physiology and chronic health evaluation II score

WBC White blood cell

PCT Procalcitonin

CRP C-reactive protein

SOFA score, Sequential Organ Failure Assessment score

IL-6 Interleukin-6

Bun Blood urea nitrogen

ALT Alanine transaminase

AST Aspartate aminotransferase

Mrp 8/14 Myeloid-related proteins 8 and 14

LUS Lung ultrasound score

ARDS Acute respiratory distress syndrome

COPD Chronic obstructive pulmonary disease

S100 A8/A9 Calprotectin or S100 calcium binding protein A8 and A9

LPS Lipopolysaccharide

ECMO Extracorporeal membrane oxygenation

COVID-19 Corona-virus disease 2019

IQR Inter quartile range

SD Standard deviation

CI Confidence interval

OR Odds ratio

ROC Receiver operating characteristic

AUC Area under the curve

Additional Information and Declarations

Competing Interests

Author Contributions

Human Ethics

Data Availability

The authors declare that they have no competing interests.

Caizhi Sun conceived and designed the experiments, performed the experiments, analyzed the data, prepared figures and/or tables, and approved the final draft.

Yongpeng Xie conceived and designed the experiments, prepared figures and/or tables, authored or reviewed drafts of the article, and approved the final draft.

Chenchen Zhu conceived and designed the experiments, analyzed the data, prepared figures and/or tables, and approved the final draft.

Lei Guo performed the experiments, analyzed the data, prepared figures and/or tables, and approved the final draft.

Bowen Xu performed the experiments, analyzed the data, prepared figures and/or tables, and approved the final draft.

Haidong Qin conceived and designed the experiments, authored or reviewed drafts of the article, and approved the final draft.

Xiaomin Li conceived and designed the experiments, authored or reviewed drafts of the article, and approved the final draft.

The following information was supplied relating to ethical approvals (i.e., approving body and any reference numbers):

This study protocol was reviewed and approved by the Institutional Review Board of Nanjing Hospital Affiliated to Nanjing Medical University (IRB no: KY20201102-03).

The following information was supplied regarding data availability:

The raw measurements are available in the Supplemental File.

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
