# Peer review of "Prognostic value of serum Mrp 8/14 in sepsis-induced acute respiratory distress syndrome patients: a retrospective cohort study"

_PeerJ, doi:10.7717/peerj.18718_

## Round 0.1 · original submission · Minor Revisions

Although most of the comments are suggestive, there are several queries too. Reviewer 1 has included an annotated file with highlighted text as well. Please address comments provided by both reviewers in a point wise manner.

·

Basic reporting

No comment

Experimental design

since its retrospective study, please clariy how consent was obtained and when patient is intubated and not able to give consent ,if the consent was taken by family and subsequently patient's consent was obtained when he/ she regained mental capacity .

Validity of the findings

- since the levels of Mrp 8/14 levels were measured on admission to ICU, they may be not necessarily the levels on the day of onset of ARDS .
-may be serial titres of Mrp8/14 on different intervals during course of ARDS could have provided better insight if the trend of the markers corellate with prognosis and mortality.
-Age group of the patients (since most of the patients are above 60 years ) you may look at the impact of age in epression of Mrp 8/14 levels as a response to ARDS.

Additional comments

88......1.The timing of measuring Mrp8/14 levels , first 24 hrs of hospitalization - this may not be the first day of onset of ARDS , hence the day of onset of ARDS and time at which the MRP 8/14 levels are sent may have an important role in validation of is levels in prognosis.
112 .....2 .informed consent was obtained from patients or family ? This point is not clear since it's a retrospective study and patients who are intubated in sedation may not be able to give consent.
251 .267...3. Serum albumin levels has lot of confounders especially in ICU patients due to extravascular shifts of fluids, variable nutrition and extent of inflammatory response. The measurement of albumin levels on admission may not be the reflection of nutritional status.

Reviewer 2 ·

Basic reporting

Thank you for the opportunity to review this very interesting manuscript by Caizhi Sun et al.
I have a few points that could be addressed in a revised version and would greatly benefit the quality of the manuscript.

The English language needs moderate editing. Please consider the following lines, although the improvements should not be limited to those lines:
17 research is reliable
22 prognostic value?
27 to the ICU
28 affiliated
29 remove "the" before patients
34-35 The area under the receiver operating characteristic curve
41 remove "the" before mortality
45-46 in the intensive care unit
46 reservoir of pathogens
57 we cannot ignore
60 "The previous clinical study": which previous clinical study? Maybe elaborate.
76 A subsequent clinical study also showed
77 and that
81 Consider using only Mrs 8/14 after explaining synonyms like S100A8/9 once.
84 our previous study has revealed
125 The LU scores (LUS stands for lung ultrasound score?)
129 remove "The"
141 was instead of were
164 remove "was"; in the survivor group
252 processes instead of procedures?
309 Additionally

The article structure is good, figures and tables are clearly described. Raw data are shared.

Please also consider describing the (patho-)physiological role and function of Mrs 8/14 in the immune system in more detail in the introduction.

Line 98: Why retrospective cohort? Blood sampling upon inclusion into the study was done prospectively, right? Maybe explain.

Experimental design

I have a few questions concerning the patient cohorts and their treatment:
1. Can you elaborate on the severity of ARDS in the cohort using the PaO2 / FiO2 ratio?
2. While ICU mortality is interesting from a clinical perspective, do you also have data on point prevalence of mortality, i.e. 30 day and 90 day mortality?
3. Can you describe the ARDS treatment of your cohort in more detail? Was ECMO used? How many patients were proned?

Validity of the findings

Overall, I find the study very interesting and worth publishing. The underlying data are provided, the statistical analysis is robust.
The conclusions are well stated and related to the original question investigated.

Please consider my suggestions above to further improve the manuscript and its readability.

---

## Round 0.2 · accepted · Accept

Authors have addressed all of the reviewers' comments and manuscript is ready for publication.

Reviewer 2 ·

Basic reporting

Thank you for providing the revised manuscript. The authors have responded to all my queries and have made significant enhancements to the manuscript. I would, however, recommend minor spelling checks in the editorial process and proofreading. I now consider the manuscript suitable for publication in the journal.

Experimental design

Research is within the aims and scopes of the journal. The research question is well defined and the methods are described with sufficient detail.

Validity of the findings

Conclusions are well stated and linked to the research question. Data have been provided.